# Measuring Faithful and Plausible Visual Grounding in VQA

**Daniel Reich, Felix Putze, Tanja Schultz**

Cognitive Systems Lab., University of Bremen, Germany

{dreich,felix.putze,tanja.schultz}@uni-bremen.de

## Abstract

Metrics for Visual Grounding (VG) in Visual Question Answering (VQA) systems primarily aim to measure a system's reliance on relevant parts of the image when inferring an answer to the given question. Lack of VG has been a common problem among state-of-the-art VQA systems and can manifest in over-reliance on irrelevant image parts or a disregard for the visual modality entirely. Although inference capabilities of VQA models are often illustrated by a few qualitative illustrations, most systems are not quantitatively assessed for their VG properties. We believe, an easily calculated criterion for meaningfully measuring a system's VG can help remedy this shortcoming, as well as add another valuable dimension to model evaluations and analysis. To this end, we propose a new VG metric that captures if a model a) identifies question-relevant objects in the scene, and b) actually relies on the information contained in the relevant objects when producing its answer, i.e., if its visual grounding is both "faithful" and "plausible". Our metric, called Faithful & Plausible Visual Grounding (FPVG), is straightforward to determine for most VQA model designs.

We give a detailed description of FPVG and evaluate several reference systems spanning various VQA architectures. Code to support the metric calculations on the GQA data set is available on GitHub[1].

## 1 Introduction

Visual Question Answering (VQA) is the task of answering natural language questions about image contents. Visual Grounding (VG) in VQA measures a VQA system's inherent proclivity to base its inference on image regions referenced in the given question and relevant to the answer. A well-grounded system infers an answer to a given question by relying on image regions relevant to the

[1] https://github.com/dreichCSL/FPVG

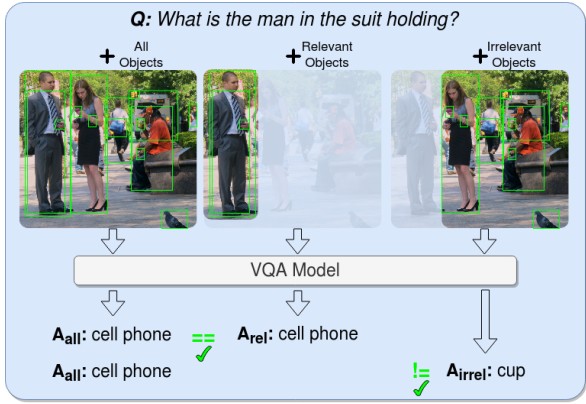

Figure 1: Faithful & Plausible Visual Grounding: The VQA model's answer given *all* objects in the image ($A_{all}$) should equal its answer when given only *relevant* objects w.r.t. the question ($A_{rel}$), and should differ when given only *irrelevant* objects ($A_{irrel}$). The figure shows a model's behavior for a question deemed faithfully and plausibly grounded.

question and plausible to humans. Hence, visually grounded inference in VQA can be broken down into two aspects: (1) Image contents impact the inference process, and (2) inference is based on *relevant* image contents. Evidence of problematic behavior that arises from a lack of (1) includes an over-reliance on language priors (Goyal et al., 2017; Agrawal et al., 2018, 2016), while a lack of (2) can cause models to react to changes in irrelevant parts of the image (Gupta et al., 2022). Both characteristics can hurt a model's capacity to provide consistent and reliable performances.

Metrics that quantify a model's VG characteristics aim to capture its internal reasoning process based on methods of model explanation. These explanations generally vary in properties of *plausibility* and *faithfulness*. *Plausible* explanations of a model's behavior prioritize human interpretability, e.g., by illustrating a clear inference path over relevant objects that lead to the decision, but might not accurately reflect a model's actual decision-making process. *Faithful* explanations, on the other hand,

prioritize a more accurate reflection of a model's decision-making process, possibly at the expense of human interpretability. Examples of plausible explanation methods are attention mechanisms (Bahdanau et al., 2014) over visual input objects, and multi-task objectives that learn to produce inference paths without conclusive involvement in the main model's answer decision (Chen et al., 2021). Faithful explanation methods may employ testing schemes with modulated visual inputs followed by comparisons of the model's output behavior across test runs (DeYoung et al., 2020; Gupta et al., 2022). While the latter types of metrics are particularly suited for the use-case of object-based visual input in VQA, they often a) require large compute budgets to evaluate the required number of input permutations (e.g. SwapMix (Gupta et al., 2022), Leave-One-Out (Li et al., 2016)); b) might evaluate in unnecessary depth, like in the case of softmax-score-based evaluations (DeYoung et al., 2020); and/or c) evaluate individual properties separately and without considering classification contexts, thereby missing the full picture (DeYoung et al., 2020; Ying et al., 2022), see also §3.4.

In this work, we propose a VG metric that is both *faithful* and *plausible* in its explanations. Faithful & Plausible Visual Grounding (FPVG) quantifies a model's faithful reliance on plausibly relevant image regions (Fig. 1). FPVG is based on a model's answering behavior for modulated sets of image input regions, similar to other faithfulness metrics (in particular DeYoung et al. (2020)), while avoiding their above-mentioned shortcomings (details in §3.4). To determine the state-of-the-art for VG in VQA, we use FPVG to measure various representative VQA methods ranging from one-step and multi-hop attention-based methods, over Transformer-based models with and without cross-modality pre-training, to (neuro-)symbolic methods. We conclude this work with investigations into the importance of VG for VQA generalization research (represented by Out-of-Distribution (OOD) testing), thereby further establishing the value of FPVG as an analytical tool. The GQA data set (Hudson and Manning, 2019) for compositional VQA is particularly suited for our tasks, as it provides detailed inference and grounding information for the majority of its questions.

**Contributions.** Summarized as follows:

- A new metric called "Faithful & Plausible Visual Grounding" (FPVG) for quantification

of plausible & faithful VG in VQA.
- Evaluations and comparisons of VQA models of various architectural designs with FPVG.
- New evidence for a connection between VG and OOD performance, provided by an empirical analysis using FPVG.
- Code to facilitate evaluations with FPVG.

## 2 Related Work

Various metrics have been proposed to measure VG in VQA models. We roughly group these into direct and indirect methods. 1) Direct methods: The most widely used methods measuring the importance of image regions to a given question are based on a model's attention mechanisms (Bahdanau et al., 2014), or use gradient-based sensitivities (in particular variants of GradCAM (Selvaraju et al., 2017)). VG is then estimated, e.g., by accumulating importance scores over matching and relevant annotated image regions (Hudson and Manning, 2019), or by some form of rank correlation (Shrestha et al., 2020). Aside from being inapplicable to non-attention-based VQA models (e.g., symbolic methods like Yi et al. (2018); Mao et al. (2019)), attention scores have the disadvantage of becoming harder to interpret the more attention layers are employed for various tasks in a model. This gets more problematic in complex Transformer-based models that have a multitude of attention layers over the input image (OSCAR (Li et al., 2020; Zhang et al., 2021), LXMERT (Tan and Bansal, 2019), MCAN (Yu et al., 2019b), MMN (Chen et al., 2021)). Additionally, attention mechanisms have been a topic of debate regarding the faithfulness of their explanation (Jain and Wallace, 2019; Wiegreffe and Pinter, 2019). Gradient-based sensitivity scores can theoretically produce faithful explanations, but require a careful choice of technique and implementation for each model individually to achieve meaningful measurements in practice (Adebayo et al., 2018; Feng et al., 2018). Various works introduce their own VG metric based on attention measurements (e.g., GQA-grounding (Hudson and Manning, 2019), VLR (Reich et al., 2022), MAC-Caps (Urooj et al., 2021)) or GradCAM-based feature sensitivities (Shrestha et al., 2020; Wu and Mooney, 2019; Selvaraju et al., 2019; Han et al., 2021). 2) Indirect methods: These include methods that measure VG based on a model's predictions under particular test (and train) conditions, e.g., with perturbations of image features (Yuan et al., 2021;

Gupta et al., 2022; Agarwal et al., 2020; DeYoung et al., 2020; Alvarez-Melis and Jaakkola, 2017), or specially designed Out-of-Distribution test sets that can inform us about a model's insufficient VG properties (Agrawal et al., 2018; Kervadec et al., 2021; Ying et al., 2022). FPVG is related to DeYoung et al. (2020) in particular and uses perturbations of image features to approximate a direct measurement of VG w.r.t. relevant objects in the input image. Thus, we categorize FPVG as an "indirect" VG evaluation method.

Finally, we note that VG can be considered a sub-problem of the VQA desiderata gathered under the term "Right for Right Reasons" (RRR) (Ross et al., 2017; Ying et al., 2022). RRR may additionally include investigations of causal behavior in a model that goes beyond (and may not be strictly dependent on) VG and may involve probing the model for its robustness and consistency in explanations, e.g., via additional (follow-up) questions (Patro et al., 2020; Selvaraju et al., 2020; Ray et al., 2019; Park et al., 2018).

## 3 Faithful & Plausible Visual Grounding

### 3.1 Metric Formulation

We propose a new metric to determine the degree of Faithful & Plausible Visual Grounding (FPVG) in a VQA model $M_{VQA}$ w.r.t. a given VQA data set $S$. Here, $S$ consists of tuples $s_j$ of question, image and answer $(q, i, a)_j$. Each such tuple in $S$ is accompanied by annotations indicating relevant regions in image $i$ that are needed to answer the question $q$. $M_{VQA}$ is characterized by its two modality inputs ($i$ and $q$) and a discrete answer output ($a$). In this paper, we expect image $i$ to be given as an object-based representation (e.g., bag of objects, scene graph) in line with the de-facto standard for VQA models[2].

FPVG requires evaluation of $M_{VQA}$ under three test conditions. Each condition differs in the set of objects representing image $i$ in each sample $s_j$ of the test. Three tests are run: 1) with all available objects ($i_{all}$), 2) with only relevant objects ($i_{rel}$), and 3) with only irrelevant objects ($i_{irrel}$). Formally, we define one dataset variant for each of these three conditions:

---

[2]In principle, FPVG can be easily adapted to work with any model (VQA or otherwise) that follows a similar input/output scheme as the standard region-based VQA models, i.e., an input consisting of N entities where a subset can be identified as "relevant" ("irrelevant") for producing a discrete output.

$$s_{j_{all}} = (q, i_{all}, a)_j, \quad s_{j_{all}} \in S_{all} \tag{1}$$

$$s_{j_{rel}} = (q, i_{rel}, a)_j, \quad s_{j_{rel}} \in S_{rel} \tag{2}$$

$$s_{j_{irrel}} = (q, i_{irrel}, a)_j, \quad s_{j_{irrel}} \in S_{irrel} \tag{3}$$

The relevance of an object in $i$ is determined by its degree of overlap with any of the objects referenced in relevance annotations for each individual question (for details, see App. A). FPVG is then calculated on a data point basis (i.e., for each question) as

$$FPVG_j = Eq(\hat{a}_{j_{all}}, \hat{a}_{j_{rel}}) \wedge \neg Eq(\hat{a}_{j_{all}}, \hat{a}_{j_{irrel}}), \tag{4}$$

where $\hat{a}_j$ is the model's predicted answer for sample $s_j$ and $Eq(x, y)$ is a function that returns True for equal answers. FPVG takes a binary value for each data point. A positive FPVG value for sample $s_{j_{all}}$ is only achieved if $M_{VQA}$'s output answers are equal between test runs with samples $s_{j_{all}}$ and $s_{j_{rel}}$, and unequal for samples $s_{j_{all}}$ and $s_{j_{irrel}}$ (reminder, that the three involved samples only differ in their visual input). The percentage of "good" (i.e., faithful & plausible) and "bad" FPVG is then given as $FPVG_+$ and $FPVG_-$, respectively:

$$FPVG_+ = \frac{1}{n} \sum_j^n FPVG_j \tag{5}$$

$$FPVG_- = 1 - FPVG_+ \tag{6}$$

We further sub-categorize FPVG to quantify correctly ($\top$) and incorrectly ($\bot$) predicted answers $\hat{a}_{j_{all}}$ as $FPVG_{\{+,-\}}^{\top}$ and $FPVG_{\{+,-\}}^{\bot}$, respectively. Hence, samples are assigned one of four categories, following their evaluation behavior (see Fig. 2 for illustration and App. B for the mathematical formulation).

### 3.2 Intuition behind FPVG

The intuition behind the object selections in $S_{rel}$ (relevant objects) and $S_{irrel}$ (irrelevant objects) is as follows:

**Testing on relevant objects $S_{rel}$.** In the context of FPVG, the output of a well-grounded system is expected to remain steady for $S_{rel}$, i.e., the model is expected to retain its original prediction from $S_{all}$, if it relies primarily on relevant visual evidence. Hence, a change in output indicates that the model has changed its focus to different visual evidence, presumably away from irrelevant features (which are dropped in $S_{rel}$) onto relevant features — a sign of "bad" grounding.

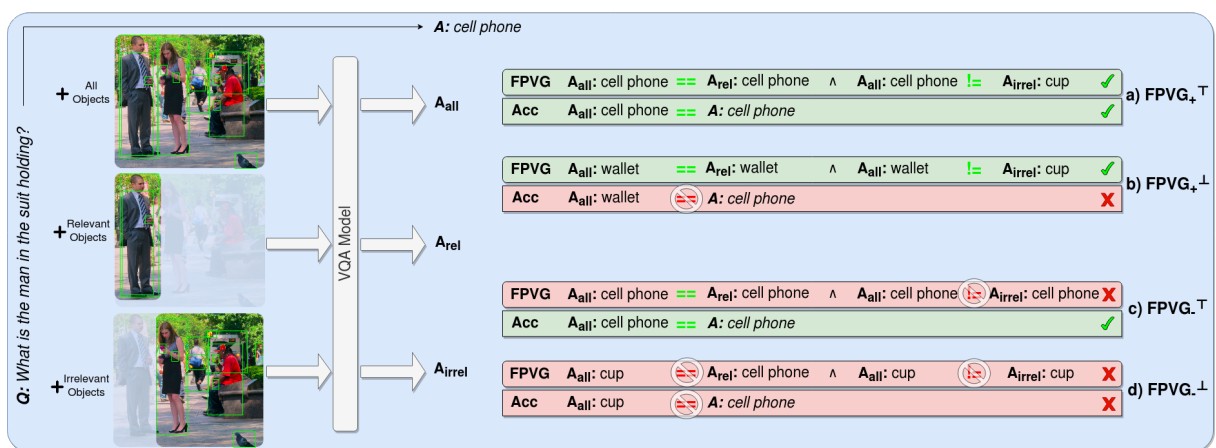

Figure 2: Examples for the four FPVG sub-categories defined in §3.1. Each sub-category encapsulates specific answering behavior for a given question in FPVG's three test cases ($A_{all}$, $A_{rel}$, $A_{irrel}$). Categorization depends on grounding status ("FPVG") and answer correctness ("Acc"). E.g., questions that return a correct answer in $A_{all}$ and $A_{rel}$ and an incorrect answer in $A_{irrel}$ are categorized as (a). The model's behavior in cases (a) and (b) satisfies the criteria for the question to be categorized as faithfully & plausibly visually grounded.

**Testing on irrelevant objects $S_{irrel}$.** In the context of FPVG, the output of a well-grounded system is expected to waver for $S_{irrel}$, i.e., the model is expected to change its original prediction in $S_{all}$, as this prediction is primarily based on relevant visual evidence which is unavailable in $S_{irrel}$.

**Summarizing expectations for well-grounded VQA.** A VQA model that relies on question-relevant objects to produce an answer (i.e., a well-grounded model that values visual evidence) should:

1. Retain its answer as long as the given visual information contains all relevant objects.
2. Change its answer when the visual information is deprived of all relevant objects and consists of irrelevant objects only.

During (1), answer flips should not happen, if the model relied only on relevant objects within the full representation $S_{all}$. However, due to tendencies in VQA models to ignore visual evidence, lack of flipping in (1) could also indicate an over-reliance on the language modality (implies indifference to the visual modality). To help rule out those cases, (2) can act as a fail-safe that confirms that a model is not indifferent to visual input[3].

The underlying mechanism can be described as an indirect measurement of the model's feature valuation of relevant objects in the regular test run $S_{all}$. The two additional experimental setups with $S_{rel}$ and $S_{irrel}$ help approximate the measurement of relevant feature valuation for $S_{all}$.

**FPVG and accuracy.** FPVG classifies samples $s_{j_{all}} \in S_{all}$ as "good" (faithful & plausible) or "bad" grounding by considering whether or not the changed visual input impacts the model's final decision, *independently of answer correctness*. Many VQA questions have multiple valid (non-annotated) answer options (e.g., "man" vs. "boy" vs. "person"), or might be answered incorrectly on account of imperfect visual features. Thus, it is reasonable to expect that questions can be well-grounded, but still produce an incorrect answer, as shown in Fig. 2, (b). Hence, FPVG categorizes samples into two main grounding categories ($FPVG_+$ and $FPVG_-$). For a more fine-grained analysis, answer correctness is considered in two additional sub-categories ($FPVG^\top$, $FPVG^\perp$) within each grounding category, as defined in Eq. 9–12.

### 3.3 Validating FPVG's Faithfulness

FPVG achieves *plausibility* by definition. In this section, we validate that FPVG's sample categorization is also driven by *faithfulness* by verifying that questions categorized as $FPVG_+$ are more faithfully grounded than questions in $FPVG_-$. To measure the degree of faithful grounding for each question, we first determine an importance ranking among the question's input objects. Then we estimate how well this ranking matches with the given relevance annotations. Three types of approaches are used in VQA to measure object importance by direct or indirect means: Measurements of a model's attention mechanism over input objects (direct), gradient-measuring methods like Grad-

---

[3]Investigations into an alternative formulation of FPVG which ignores requirement (2) can be found in App. C.

| Method | relevant | | irrelevant | |
|---|---|---|---|---|
| | $FPVG_+\uparrow$ | $FPVG_-\downarrow$ | $FPVG_+\downarrow$ | $FPVG_-\uparrow$ |
| Attention | 60.9 | 26.6 | 16.7 | 51.2 |
| GradCAM | 10.4 | 8.5 | 53.7 | 67.4 |
| LOO | 29.8 | 16.0 | 52.0 | 71.7 |

Table 1: Ranking match percentage between feature importance rankings and relevant/irrelevant objects for questions in $FPVG_+$ and $FPVG_-$. Model: UpDn.

CAM (direct), and methods involving input feature manipulations followed by investigations into the model's output change (indirect).
VQA-model UpDn's (Anderson et al., 2018) attention and the feature manipulation method Leave-One-Out (LOO[4]) (Li et al., 2016) were found to deliver the most faithful measurements of feature importance in experiments with UpDn on GQA in Ying et al. (2022). We use these two methods and also include GradCAM used in Selvaraju et al. (2019); Shrestha et al. (2020) for completeness.

We measure UpDn's behavior on GQA's balanced validation set (see §4.1). Table 1 lists the ranking match degree between object importance rankings (based on $S_{all}$) and relevance annotations, averaged over questions categorized as $FPVG_+$ and $FPVG_-$, respectively. The "relevant" ("irrelevant") category produces a high score if all relevant (irrelevant) objects are top-ranked by the used method (see App. D.2 for details). Hence, faithfully grounded questions are expected to score highly in the "relevant" category, as relevant objects would be more influential to the model's decision.

Results show that object importance rankings over the same set of questions and model vary greatly across methods. Nonetheless, we find that data points in both $FPVG_+$ and $FPVG_-$ achieve on avg favorable scores across all three metrics with mostly considerable gaps between opposing categories (i.e., $+$ and $-$). This is in line with expectations and confirms that FPVG's data point categorization is driven by faithfulness.

### 3.4 Comparison with "sufficiency" and "comprehensiveness"

Two metrics to measure faithfulness in a model, "sufficiency" and "comprehensiveness", were proposed in DeYoung et al. (2020) and used in the context of VQA in similar form in Ying et al. (2022).

---
[4]LOO evaluates a model N times (N=number of input objects), each time "leaving-out-one object" of the input and observing the original answer's score changes. A large score drop signifies high importance of the omitted object.

"Sufficiency" and "comprehensiveness" are similar to FPVG and therefore deserve a more detailed comparison. They are calculated as follows.

**Definition.** Let a model $M_\theta$'s answer output layer be represented as softmax-normalized logits. A probability distribution over all possible answers is then given as $p(a|q, i_{all}) = m_\theta(q, i_{all})$. The max element in this distribution is $M_\theta$'s predicted answer, i.e., $\hat{a} = \underset{a}{argmax}\, p(a|q, i_{all})$, where the probability for the predicted answer is given by $p_{\hat{a}_{all}} = M_\theta(q, i_{all})_{\hat{a}}$.

**Sufficiency** is defined as the change of output probability of the predicted class given *all* objects vs. the probability of that same class given only *relevant* objects:

$$suff = p_{\hat{a}_{all}} - p_{\hat{a}_{rel}} \qquad (7)$$

**Comprehensiveness** is defined as the change of output probability of the predicted class given *all* objects vs. the probability of that same class given only *irrelevant* objects:

$$comp = p_{\hat{a}_{all}} - p_{\hat{a}_{irrel}} \qquad (8)$$

A faithfully grounded model is expected to achieve low values in $suff$ and high values in $comp$.

**Object relevance and plausibility.** The definition of what constitutes relevant or irrelevant objects is crucial to the underlying meaning of these two metrics. FPVG uses annotation-driven object relevance discovery and subsequently determines a model's faithfulness w.r.t. these objects. Meanwhile, Ying et al. (2022) estimates both metrics using *model-based* object relevance rankings (e.g., using LOO), hence, measuring the degree of faithfulness a model has towards model-based valuation of objects as determined by an object importance metric. A separate step is then needed to examine these explanations for "plausibility". In contrast, FPVG already incorporates this step in its formulation, which determines if the model's inference is similar to that of a human by measuring the degree of *faithful* reliance on *plausibly* relevant objects (as defined in annotations).

**Advantages of FPVG.** FPVG overcomes the following shortcomings of $suff$ and $comp$:

1. $Suff$ and $comp$ are calculated as an average over the data set independently of each other and therefore do not evaluate the model for presence of *both* properties in each data point.

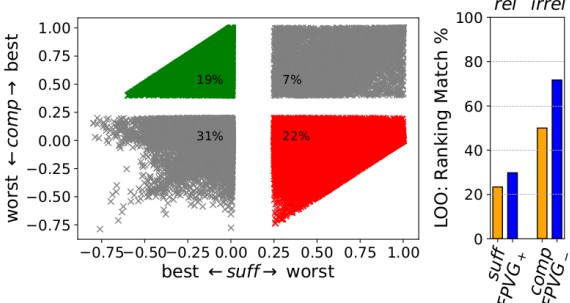 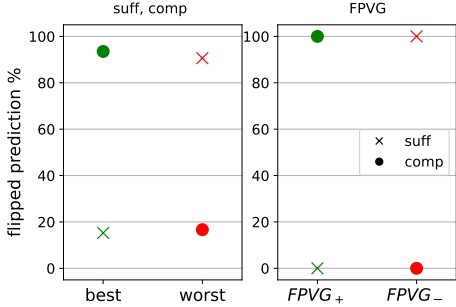

Figure 3: Left: Percentage of samples with best (worst) $suff$ & $comp$ scores (medium scores not pictured). Many samples with the $suff$ property lack $comp$ and vice-versa (gray). Right: LOO-based ranking match percentages for samples in $suff$, $comp$ and FPVG (higher is better). Model: UpDn.

Figure 4: Sample distribution and answer class flip percentages depending on metric categorization. X-axis: VG quality categories based on $suff$ & $comp$ (left) and FPVG (right). Y-axis: percentage of flipped answers in each category. Note that in this figure, FPVG's formulation is interpreted in terms of $suff$ (Eq. 4, right side, left term) and $comp$ (right term). Model: UpDn.

2. $Suff$ and $comp$ only consider prediction probabilities of the maximum class in isolation, which means that even a change in model output as significant as a flip to another class may be declared insignificant by these metrics (e.g., for $suff$, if the output distribution's max probability $p_{\hat{a}_{all}}$ is similar to $p_{\hat{a}_{rel}}$).

**Shortcoming 1.** Fig. 3, left, illustrates why isolating the two properties can cause inaccurate readings (1). The analyzed model assigns "good" $suff$ scores (defined in Ying et al. (2022) as $< 1\%$ abs. prob. reduction from $p_{\hat{a}_{all}}$ to $p_{\hat{a}_{rel}}$) to a large number of questions (left two quadrants in Fig. 3, left). However, many of these questions also show "bad" $comp$ ($< 20\%$ abs. drop from $p_{\hat{a}_{all}}$ to $p_{\hat{a}_{irrel}}$) (lower left quadrant in Fig. 3, left), which reflects model behavior that one might observe when visual input is ignored entirely. Thus, the full picture is only revealed when considering both properties in conjunction, which FPVG does. Further evidence of the drawback stemming from (1) is pictured in Fig. 3, right, which shows avg LOO-based ranking match percentages (cf. §3.3) for data points categorized as "best" $suff$ or $comp$ and FPVG. Data points in FPVG's categories score more favorably than those in $suff$ and $comp$, illustrating a more accurate categorization.

**Shortcoming 2.** Fig. 4, left, illustrates problem (2). A large percentage of questions with best (=low) scores in $suff$ flip their answer class (i.e., fail to reach 0% flipped percentage), even when experiencing only minimal class prob drops ($< 1\%$ abs.). Similarly, some percentage of questions with best (=high) $comp$ scores fail to flip their answer (i.e., fail to reach 100% flipped percentage),

even though the class prob dropped significantly ($>= 40\%$ abs. drop). Both described cases show that failure to consider class probs in the context of the full answer class distribution negatively impacts the metric's quantification of a model's VG capabilities w.r.t. actual effects on its answer output behavior. FPVG's categorization avoids this issue by being defined over actual answer changes (Fig. 4, right: flipped prediction percentages per VG category are always at the expected extremes, i.e., 0% or 100%).

**Summary.** FPVG avoids shortcoming (1) by taking both $suff$ and $comp$ into account in its joint formulation at the data point level, and (2) by looking at actual answer output changes (Fig. 4, right) and thus implicitly considering class probs over all classes and employing meaningful decision boundaries for categorization. Additionally, relying on answer flips instead of an abstract softmax score makes FPVG more intuitively interpretable.

### 3.5 Discussion on other existing metrics

FPVG relies on the method of feature deletions to determine "faithful" reliance on a "plausible" set of inputs. Other VG metrics exist that instead rely on GradCAM (Shrestha et al., 2020) or a model's Attention mechanism (Hudson and Manning, 2019) to provide a "faithful" measurement of input feature importance (see also App. D.1). The two mentioned metrics leverage these measurements to determine if a model relies on "plausibly" relevant objects. For instance, Shrestha et al. (2020) calculates a ranking correlation between the measured GradCAM scores and the rankings based on (plausible) object relevance annotations. The metric in

Hudson and Manning (2019) sums all of a model's Attention values assigned to visual input objects that have been determined to represent plausible objects.

While "plausibility" is straightforwardly achieved by appropriate selection of plausibly relevant reference objects (which would be the same across these metrics), the property of "faithfulness" is more difficult to obtain and heavily dependent on the employed feature importance technique. Investigations in Ying et al. (2022) cast doubt on the faithfulness of GradCAM measurements, with feature deletion techniques and Attention mechanism scoring most favorably in faithfulness in the explored setting. However, as discussed in §2, the faithfulness of Attention measurements has not been without scrutiny, and is not straightforward to extract correctly in models that make heavy use of Attention mechanisms (such as Transformers). Based on this evidence, we find the method of feature deletions to be the most sensible and versatile choice to achieve faithfulness of measurements in FPVG across a wide range of model architectures in VQA.

## 4 Experiments

### 4.1 Preliminaries

The GQA data set Hudson and Manning (2019) provides detailed grounding information in available train & validation sets. Contrary to HAT (Das et al., 2016), which consists of human attention data for a small percentage of questions in the VQA data set (Goyal et al., 2017), GQA contains automatically generated relevance annotations for most questions in the dataset. Our experiments focus on GQA, but FPVG can theoretically be measured with any VQA data set containing the necessary annotations, like HAT. In this work, we rely on GQA's "balanced" split (943k samples), but use the full train split (14m samples) for some models if required in their official training instructions. Testing is performed on the balanced val set (132k samples).

Details regarding object relevance determination and model training can be found in App. A and E.

### 4.2 Models

To provide a broad range of reference evaluations with FPVG, we evaluate a wide variety of model designs from recent years: UpDn (Anderson et al., 2018) is an attention-based model that popularized the contemporary standard of object-based image representation. MAC (Hudson and Manning, 2018) is a multi-hop attention model for multi-step inference, well-suited for visual reasoning scenarios like GQA. MCAN (Yu et al., 2019b), MMN (Chen et al., 2021) and OSCAR+ (Zhang et al., 2021) are all Transformer-based (Vaswani et al., 2017) models. MMN employs a modular design that disentangles inference over the image information from the question-based prediction of inference steps as a functional program in a separate process, thereby improving interpretability compared to monolithic systems like MCAN. MMN also makes an effort to learn correct grounding using an auxiliary loss. OSCAR+ uses large-scale pre-training on multiple V+L data sets and is subsequently fine-tuned on GQA's balanced train set. We use the official release of the pre-trained OSCAR+ base model (which uses proprietary visual features) and fine-tune it. DFOL (Amizadeh et al., 2020) is a neuro-symbolic method that disentangles vision from language processing via a separate question parser similar to MMN and VLR (Reich et al., 2022). The latter is a modular, symbolic method that prioritizes strong VG over accuracy by following a retrieval-based design paradigm instead of the commonly employed classification-based design in VQA.

In addition to these main models, we include two methods that focus on grounding improvements and are both applied to UpDn model training: HINT (Selvaraju et al., 2019) aligns GradCAM-based (Selvaraju et al., 2017) feature sensitivities with annotated object relevance scores. VisFIS (Ying et al., 2022) adds an ensemble of various RRR/VG-related objective functions (including some data augmentation) to the training process.

All models, except for OSCAR+, were trained using the same 1024-dim visual features generated by a Faster R-CNN object detector trained on images in GQA using Detectron2 (Wu et al., 2019).

### 4.3 Evaluations

Results are listed in Table 2, sorted by $FPVG_+$ (last column). Our first observation is that FPVG and accuracy are not indicative of one another, confirming that our metric for grounding is complementary to accuracy and adds a valuable dimension to VQA model analysis. Secondly, we see that (neuro-)symbolic methods like DFOL, and VLR in particular, stand out among (non-VG-boosted) VQA models in terms of FPVG, even while trailing in accuracy considerably. Thirdly, we find that

| Model | Obj. Det. | Acc | $Acc_{all}$ | $Acc_{rel}\uparrow$ | $Acc_{irrel}\downarrow$ | $FPVG_+^\top\uparrow$ | $FPVG_+^\perp$ | $FPVG_-^\top\downarrow$ | $FPVG_-^\perp$ | $FPVG_+\uparrow$ |
|---|---|---|---|---|---|---|---|---|---|---|
| MAC (Hudson and Manning, 2018) | Det2 | 60.23 | 59.20 | 58.12 | 44.33 | 15.40 | 7.19 | 43.81 | 33.60 | 22.59 |
| UpDn (Anderson et al., 2018) | Det2 | 55.53 | 57.99 | 58.51 | 44.32 | 15.76 | 9.68 | 42.23 | 32.33 | 25.44 |
| UpDn+HINT (Selvaraju et al., 2019) | Det2 | 55.56 | 57.95 | 57.88 | 42.98 | 16.31 | 9.72 | 41.64 | 32.33 | 26.03 |
| MCAN (Yu et al., 2019b) | Det2 | 66.18 | 65.78 | 67.3 | 44.62 | 20.18 | 6.20 | 45.60 | 28.02 | 26.37 |
| OSCAR+ (Zhang et al., 2021) | VinVL | 70.52 | **69.96** | **71.79** | 50.24 | 20.37 | 6.00 | 49.58 | 24.05 | 26.37 |
| MMN (Chen et al., 2021) | Det2 | 68.49 | 68.23 | 64.37 | 43.93 | 21.93 | 5.86 | 46.29 | 25.92 | 28.22 |
| DFOL (Amizadeh et al., 2020) | Det2 | 55.79 | 57.45 | 57.36 | 36.70 | 20.19 | 10.03 | 37.25 | 32.53 | 30.22 |
| UpDn+VisFIS (Ying et al., 2022) | Det2 | 57.09 | 60.01 | 63.71 | 43.25 | 20.38 | 12.20 | 39.63 | 27.79 | 32.58 |
| VLR (Reich et al., 2022) | Det2 | 57.25 | 57.39 | 61.29 | **35.99** | **24.55** | 11.68 | **32.83** | 30.93 | **36.23** |
| UpDn* (Anderson et al., 2018) | VinVL | 65.22 | 64.81 | 68.28 | 43.00 | 23.90 | 9.29 | 40.92 | 25.89 | 33.19 |

Table 2: FPVG results for various models, sorted by $FPVG_+$. Accuracy (Acc) is calculated on GQA balanced val set, while all others are calculated on a subset (see App. Table 5 for size). Blue arrows show desirable behavior for well-grounded VQA in each category[5](best results in bold). Last line: Results for UpDn* trained with VinVL features are included to allow an easier assessment of OSCAR+ (w/ VinVL) results.

| | Accuracy | | $FPVG_+$ | |
|---|---|---|---|---|
| Model | ID | OOD | ID | OOD |
| UpDn | 51.4±.58 | 30.83±1.96 | 17.5±.87 | 19.33±.73 |
| HINT | 51.28±.39 | 31.34±.55 | 18.06±1.23 | 19.59±.68 |
| VisFIS | 53.28±.44 | 33.42±1.03 | 25.1±.78 | 25.18±.94 |
| MAC | 52.1±.46 | 31.31±.5 | 15.4±.51 | 16.72±.22 |
| MMN | 52.28±.43 | 36.48±.56 | 18.74±.32 | 17.88±.6 |
| VLR | 55.64 | 56.38 | 37.56 | 38.51 |

Table 3: Accuracy (i.e., $Acc_{all}$) and $FPVG_+$ for models evaluated with GQA-101k over five differently seeded training runs.

methods that boost grounding characteristics, like VisFIS, show promise for closing the gap to symbolic methods - if not exceeding them. Lastly, we observe that $FPVG_+$ is generally low in all evaluated models, indicating that there is still ample room for VG improvements in VQA.

## 4.4 Connection to Out-of-Distribution (OOD)

We use FPVG to gain insights into the challenge of OOD settings by analyzing VQA models with GQA-101k (Ying et al., 2022), a dataset proposed for OOD testing. GQA-101k consists of a repartitioned train/test set based on balanced GQA and was created following a similar methodology as the OOD split called VQA-CP (Agrawal et al., 2018).

Results in Table 3 show median values and maximum deviation thereof over five differently seeded training runs per model type (note that VLR uses deterministic inference, so no additional runs were performed for it). Table 4 lists correct-to-incorrect (c2i) answer ratios for six model types trained and evaluated on GQA-101k. The c2i ratios are determined for each test set (ID/OOD) and $FPVG_{\{+,-\}}$. They are calculated as number of correct answers divided by number of incorrect

[5]FPVG sub-categories $FPVG_+^\perp$ and $FPVG_-^\top$ have no intuitively sensible ranking directive under the FPVG motivation.

answers, hence, a c2i ratio of $> 1$ reflects that correct answers dominate the considered subset of test questions. In the following analysis, we leverage the listed c2i ratios to investigate and illustrate the connection between VG and (OOD) accuracy.

### 4.4.1 Understanding the connection between FPVG and accuracy.

In Table 2 and 3 we observe a somewhat unpredictable relationship between $FPVG_+$ and accuracy. We analyze the c2i ratios in Table 4 to gain a better understanding of this behavior. Table 4 shows that FPVG-curated c2i ratios can vary substantially across model types (e.g., UpDn vs. MMN). These ratios can be interpreted as indicators of how effectively a model can handle and benefit from correct grounding. Large differences between models' c2i profiles explain why the impact of VG on accuracy can vary significantly across models. E.g., MMN has a much stronger c2i profile than UpDn, which explains its higher OOD accuracy even with lower $FPVG_+$.

### 4.4.2 Understanding the connection between FPVG and OOD performance.

The inter-dependency of VG and OOD performance plays an important role in VQA generalization. FPVG can help us gain a deeper understanding.

**More OOD errors when VG is bad.** Fig. 5, left, depicts relative c2i ratio degradation when comparing ID to OOD settings. All models suffer a much higher c2i drop for questions categorized as $FPVG_-$ than $FPVG_+$. In other words, models make more mistakes in an OOD setting in general, but they tend to do so *in particular when questions are not correctly grounded*. Note, that VLR is affected to a much lower degree due to its quasi-insensitivity to Q/A priors.

| | $FPVG_+$ | | $FPVG_-$ | |
|---|---|---|---|---|
| Model | ID | OOD | ID | OOD |
| UpDn | 1.35±.09 | .77±.05 | 1.11±.03 | .43±.04 |
| HINT | 1.36±.07 | .85±.05 | 1.11±.02 | .43±.01 |
| VisFIS | 1.4±.06 | .84±.06 | 1.23±.02 | .47±.02 |
| MAC | 1.44±.06 | .77±.05 | 1.16±.02 | .45±.01 |
| MMN | 1.91±.05 | 1.21±.12 | 1.11±.02 | .57±.02 |
| VLR | 1.91 | 2.12 | 1.09 | 1.05 |

Table 4: Correct to incorrect (c2i) answer ratios for questions categorized as $FPVG_{\{+,-\}}$. Data set: GQA-101k.

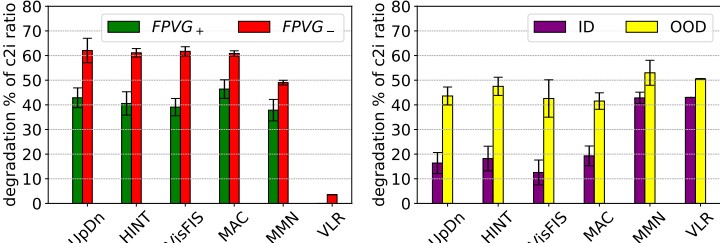

Figure 5: Performance drops when comparing ID to OOD (questions in $FPVG_{\{+,-\}}$, left), and when comparing $FPVG_+$ to $FPVG_-$ (questions in ID/OOD, right). Data set: GQA-101k.

**VG is more important to OOD than ID.** Fig. 5, right, shows accuracy sensitivity towards changes in grounding quality, i.e., when comparing $FPVG_+$ to $FPVG_-$. We draw two conclusions: 1) All models suffer from c2i degradation, hence, they all tend to make more mistakes for questions categorized as $FPVG_-$ than $FPVG_+$. 2) This tendency is (considerably) more pronounced in OOD which provides evidence that *OOD performance is particularly sensitive to grounding*.

**Summary.** Our analysis shows that *VQA models have a clear tendency to make mistakes in OOD for questions that are not faithfully grounded*. This tendency is consistently observed across various model types and model instances. Our findings support the idea that weak visual grounding is detrimental to accuracy in OOD scenarios in particular, where the model is unable to fall back on learned Q/A priors to find the correct answer (as it can do in ID testing). Furthermore, we note that VisFIS, which boasts considerable improvements in FPVG and strong improvements in accuracy over basic UpDn, is unable to overcome these problematic tendencies. This suggests that VG-boosting methods *alone* might not be enough to overcome a model's fixation on language-based priors, which is exacerbating the performance gap between ID/OOD.

## 5   Conclusion

We introduced Faithful & Plausible Visual Grounding (FPVG), a metric that facilitates and streamlines the analysis of VG in VQA systems. Using FPVG, we investigated VQA systems of various architectural designs and found that many models struggle to reach the level of faithful & plausible VG that systems based on symbolic inference can provide. Finally, we have shown that FPVG can be a valuable tool in analyzing VQA system behavior, as exemplified by investigations of the VG-OOD relationship. Here, we found that VG plays an important role in OOD scenarios, where, compared to ID scenarios, bad VG leads to considerably more errors than good VG, thus providing us with a compelling argument for pursuing better-grounded models.

## 6   Limitations

Plausibility of explanations in FPVG is assumed to be provided by accurate, unambiguous and complete annotations of relevant objects per evaluated question. Although the GQA data set provides annotations in the shape of relevant object pointers during the inference process for a question, these annotations may be ambiguous or incomplete. For instance, a question about the color of a soccer player's jersey might list pointers to a single player in an image where multiple players are present. Excluding only this one player from the image input based on the annotated pointer would still include other players (with the same jersey) for the $S_{irrel}$ test case. In such cases, FPVG's assumptions would be violated and its result rendered inaccurate. In this context, we also note that FPVG's behavior has not been explicitly explored for cases with ambiguous relevance annotations.

Secondly, FPVG creates its visual input modulations by matching annotated objects with objects detected by an object detector. Different object detectors can produce bounding boxes of varying accuracy and quantity depending on their settings. When using a new object detector as a source for visual features, it might be necessary to re-adjust parameters used for identifying relevant/irrelevant objects (see App. A for settings used in this work). When doing so, the integrity of FPVG can only be retained when making sure that there are no overlapping objects among relevant & irrelevant sets.

Thirdly, comparing VQA models with FPVG across visual features produced by different object

detectors might be problematic/inaccurate in itself, as 1) different numbers of objects are selected for relevant & irrelevant sets, and 2) different Q/A samples might be evaluated (e.g., due to missing detections of any relevant objects). If possible, when using a new object detector, we recommend including FPVG evaluations for some reference model(s) (e.g., UpDn) as an additional baseline to enable an improved assessment of a model's FPVG measurements that are trained with a different object detector's visual features.

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

## A  Determining relevant objects.

FPVG can only be meaningfully evaluated with questions for which the used object detector found both relevant and irrelevant objects. If e.g. no question-relevant objects were detected, the question is excluded. Hence, different subsets of the test (=balanced val set) are evaluated depending on the used object detector. Table 5 lists some statistics related to this for each of the object detectors used in our evaluations. The set of relevant objects is determined by $IoU > 0.5$ between detected & annotated bbox. The set of irrelevant objects excludes all detected bboxes that cover $> 25\%$ of any annotated relevant object to avoid any significant inclusion of relevant image content.

| Obj. Detector | #Q/A | #obj max | #obj avg all \| rel \| irrel |
|---|---|---|---|
| Detectron2 (Wu et al., 2019) | 114k | 100 | 91 \| 5 \| 62 |
| VinVL (Zhang et al., 2021) | 110k | 100 | 45 \| 2 \| 31 |

Table 5: Object detector bbox statistics for FPVG evaluation.

## B  Metric formulation - addendum

Mathematical formulation of each of FPVG's four subcategories is as follows (for a description of the variables used in the formulae, see §3.1):

$$FPVG_+^\top = \tfrac{1}{n} \sum_j^n (FPVG_j * Eq(\hat{a}_{j_{all}}, a_j)) \quad (9)$$

$$FPVG_+^\bot = \tfrac{1}{n} \sum_j^n (FPVG_j * (1 - Eq(\hat{a}_{j_{all}}, a_j))) \quad (10)$$

$$FPVG_-^\top = \tfrac{1}{n} \sum_j^n ((1 - FPVG_j) * Eq(\hat{a}_{j_{all}}, a_j)) \quad (11)$$

$$FPVG_-^\bot = \tfrac{1}{n} \sum_j^n ((1 - FPVG_j) * (1 - Eq(\hat{a}_{j_{all}}, a_j))) \quad (12)$$

Eq. 9–12 sum to 1. See Fig. 2 for illustration, where image-to-equation correspondence is given by (a)-Eq. 9, (b)-Eq. 10, (c)-Eq. 11, (d)-Eq. 12.

## C  Metric investigations - modified FPVG

During the paper review of this work, investigations were requested to show the value of FPVG's third test case (which involves irrelevant objects and is run to acquire $A_{irrel}$) by exploring FPVG's behavior when $A_{irrel}$ is omitted. We include our findings here. Note, that this section assumes that the reader has read the main paper.

**Theoretical considerations.** One motivation for considering an answer change when testing with irrelevant parts is given in §3.2, namely that it uncovers cases where the model is simply indifferent to visual input entirely. This indifference to visual input is a major (language) bias problem in VQA. Hence, it is important to have a mechanism that can identify these cases.

**Empirical investigation.** We investigate results produced by the modified FPVG version. We modify FPVG to only consider answer changes when testing with relevant objects (i.e., ignoring the third test involving irrelevant objects and therefore removing the condition for $A_{irrel}$ from FPVG's formulation). Results of this modified FPVG metric ($mod\_FPVG$) for ID/OOD tests over five runs (same tests that were discussed in §4.4) are shown in Table 6.

**Discussion.** Results of $mod\_FPVG$ appear to be less reasonable than the original FPVG. E.g., VLR, which achieved by far the best $FPVG_+$ (and OOD accuracy) with the original FPVG, is now ranked behind VisFIS and close to UpDn. MMN, which had the best OOD performance among classification-based models (and was ranked third in $FPVG_+$) is now ranked last by a large margin. Based on the known architectural properties of these models (e.g., using VG-focused mechanisms in MMN and VLR), such rankings would be surprising.

We also investigate the c2i ratios for $mod\_FPVG$ in the same scenario (see Table 7 and Fig. 6). Here, we observe opposite trends to the ones shown in the main paper for original FPVG. In particular, these new results suggest that well-grounded questions (as per $mod\_FPVG$) are much more prone to producing wrong answers in OOD vs. ID than badly-grounded questions (as illustrated by larger degradations for $mod\_FPVG_+$ than $mod\_FPVG_-$ in Fig. 6, left). This does not align with any reasonable expectation for a model's OOD behavior and we think it again points to problems with the modified metric.

**Conclusion.** These empirical results on top of the mentioned theoretical considerations emphasize the value of including tests with irrelevant objects in FPVG.

| Model | Accuracy | | $FPVG_+$ | | $mod\_FPVG_+$ | |
|---|---|---|---|---|---|---|
| | ID | OOD | ID | OOD | ID | OOD |
| UpDn | 51.4±.58 | 30.83±1.96 | 17.5±.87 | 19.33±.73 | 77.56±1.43 | 76.33±1.23 |
| HINT | 51.28±.39 | 31.34±.55 | 18.06±1.23 | 19.59±.68 | 74.81±1.62 | 75.37±2.80 |
| VisFIS | 53.28±.44 | 33.42±1.03 | 25.1±.78 | 25.18±.94 | 82.25±0.40 | 80.15±0.91 |
| MAC | 52.1±.46 | 31.31±.5 | 15.4±.51 | 16.72±.22 | 73.93±2.08 | 73.27±2.56 |
| MMN | 52.28±.43 | 36.48±.56 | 18.74±.32 | 17.88±.6 | 61.99±0.62 | 58.66±1.15 |
| VLR | 55.64 | 56.38 | 37.56 | 38.51 | 79.48 | 79.03 |

Table 6: Accuracy (i.e., $Acc_{all}$) and $mod\_FPVG_+$ for models evaluated with GQA-101k over five differently seeded training runs.

| Model | $mod\_FPVG_+$ | | $mod\_FPVG_-$ | |
|---|---|---|---|---|
| | ID | OOD | ID | OOD |
| UpDn | 1.62±.06 | .59±.08 | .35±.03 | .22±.01 |
| HINT | 1.68±.05 | .61±.03 | .38±.03 | .24±.03 |
| VisFIS | 1.75±.05 | .70±.03 | .24±.01 | .15±.02 |
| MAC | 1.74±.07 | .61±.03 | .42±.04 | .29±.02 |
| MMN | 2.29±.06 | .98±.06 | .46±.01 | .33±.02 |
| VLR | 2.01 | 2.10 | .25 | .24 |

Table 7: Correct to incorrect (c2i) answer ratios for questions categorized as $mod\_FPVG_{\{+,-\}}$. Data set: GQA-101k.

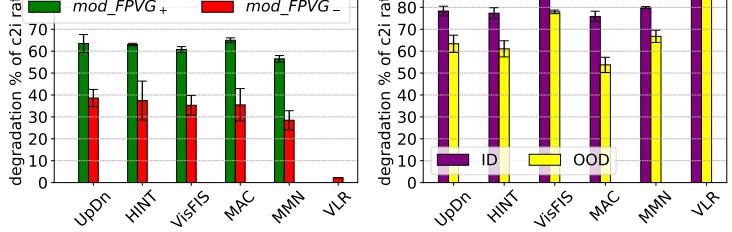

Figure 6: Performance drops when comparing ID to OOD (questions in $mod\_FPVG_{\{+,-\}}$, left), and when comparing $mod\_FPVG_+$ to $mod\_FPVG_-$ (questions in ID/OOD, right). Data set: GQA-101k.

## D  Feature importance

### D.1  Methods for measuring feature importance

In 3.3, we consider three methods to measure feature importance, one representative from each of the three categories commonly used in VQA, described in more detail in the following:

1. Measuring attention (direct): Attention over input objects gives a sense of importance the model assigns to each object (used, e.g., in Li et al. (2019); Urooj et al. (2021); Hudson and Manning (2019)).

2. Measuring gradients (direct): Gradient-based methods like GradCAM are close to the model's inner workings as they involve estimating a direct link between the importance of the input features and a model's output decision (used, e.g., in Selvaraju et al. (2019); Wu and Mooney (2019)).

3. Feature manipulation (indirect): Usually by omission of input entities (i.e., vectors representing objects). The manipulated image representation can be zero-padded to maintain the model's size expectations, as is commonly done for variable length inputs in sequence modeling. Other variants used in VQA

include replacing omitted objects with certain other values (e.g., constants (Ying et al., 2022), objects from other images (Yuan et al., 2021; Gupta et al., 2022)).

### D.2  Feature importance ranking scores

Scores in Table 1 were calculated as follows:

A question's "relevant" score measures how many of $N$ annotated relevant objects in set $relN$ are among the $topN$ relevant objects (as determined and ranked by the used metric). It is calculated as $\frac{topN \cap relN}{relN}$, where a higher value is desirable for $FPVG_+$). A question's "irrelevant" score measures how many of $M$ annotation-determined irrelevant objects in set $irrelM$ are among the $topM$ metric-determined relevant objects. It is calculated as $\frac{topM \cap irrelM}{irrelM}$, with a lower value being desirable for $FPVG_+$.

## E  Model Training

In this section we include details for training procedures of models used in this work's evaluations. Generally, we use GQA's balanced train set to train all models and the balanced val set for evaluations. A small dev set (either a small, randomly excluded partition of the train set (20k questions), or separately provided in case of experiments on GQA-

101k (Ying et al., 2022)) is used for model selection.

### E.0.1 Visual Features

The object detector used in this work is a Faster R-CNN (Ren et al., 2015) model with ResNet101 (He et al., 2016) backbone and an FPN (Lin et al., 2017) for region proposals. We trained this model using Facebook's Detectron2 framework (Wu et al., 2019). The ResNet101 backbone model was pretrained on ImageNet (Deng et al., 2009).

The object detector was trained for GQA's 1702 object classes using 75k training images (images in GQA's train partition). Training lasted for 1m iterations with mini-batch of 4 images, using a multi-step learning rate starting at 0.005, reducing it by a factor of 10 at 700k and again at 900k iterations. No other parameters were changed in the official Detectron2 training recipe for this model architecture. Training took about 7 days on an RTX 2080 Ti.

We extract 1024-dim object-based visual features from a layer in the object classification head of this model which acts as input to the final fully-connected softmax-activated output layer. Up to 100 objects per image are selected as follows: per-class NMS is applied at 0.7 IoU for objects that have any softmax object class probability of $> 0.05$.

Note that with exception of GQA-101k's repartitioned test sets (which mix questions from balanced train and val sets), no images used in testing were used in training.

Most models are trained with Detectron2-based visual features (1024-dim object-based visual features for 100 objects/image max) as input. For OSCAR+, we use the officially released pre-trained base model which uses VinVL visual features (Zhang et al., 2021).

### E.0.2 MMN

MMN (Chen et al., 2021) consists of two main modules that are trained separately: A program parser and the actual inference model, which takes the predicted program from the parser as input. We mostly follow the settings in the official code-base but detail some aspects of our customization here.

For the inference model, we run up to 5 epochs of bootstrapping (using GQA's "all" train set (14m questions)) with Oracle programs and another up to 12 epochs of fine-tuning with parser-generated programs (from the official release), using GQA's

balanced train set (1m questions). We use early stopping of 1 epoch and select the model by best accuracy on the dev set (using Oracle programs in bootstrapping mode and predicted programs in fine-tuning mode). The program parser was not retrained.

### E.0.3 DFOL

DFOL (Amizadeh et al., 2020) uses a vanilla seq2seq program parser, but neither code nor generated output for this is provided in the official code base. Thus, evaluations are run with ground-truth programs from GQA. DFOL is trained on a loss based on answer performance to learn weights in its visual encoding layers that produce an image representation similar to the one used by VLR (Reich et al., 2022), given high-dimensional visual features as input.

Training is done based on the official instructions for a complex 5-step curriculum training procedure. We train the first 4 curriculum steps with the entire 14 million questions in GQA's "all" training data partition, as specified in the instructions. As this is extremely resource intensive, we train for one epoch in each step. Finally, we run the 5th step with the "balanced" train data only ( 1m questions) until training finishes by early stopping of 1 epoch.

### E.0.4 MAC

MAC (Hudson and Manning, 2018) is a monolithic VQA model based on a recurrent NN architecture which allows specification of the number of inference steps to take over the knowledge base. We follow the official training procedure guidelines given in the released code base and use 4-step inference. We train the model on GQA's balanced train set and use early stopping of 1 epoch based on accuracy on a dev set to select the best model.

### E.0.5 UpDn, HINT, VisFIS

UpDn (Anderson et al., 2018) is a classic, straight-forward attention-based model with a single attention step before merging vision and language modalities. We use the implementation shared by (Ying et al., 2022). Following the scripts there, we train UpDn for 50 epochs and select the best model based on accuracy on a dev set.

HINT (Selvaraju et al., 2019) and VisFIS (Ying et al., 2022) are two VG-improvement methods. VisFIS is trained according to the released scripts. HINT is trained according to Shrestha et al. (2020) (using the VisFIS codebase), i.e. we continue train-

ing the baseline UpDn model with HINT (using GQA annotations to determine importance scores) for 12 more epochs and select the best resulting model (accuracy on dev set).

### E.0.6 VLR

VLR (Reich et al., 2022) is a modular, symbolic method that requires a full scene graph as visual representation. Similar to DFOL and MMN, it makes use of a (trained) program parser. The actual inference module does not require training. Training of the program parser and generation of the scene graph was done according to the description in Reich et al. (2022). The scene graph was generated using the same Detectron2 model that produced the visual features for the other models in this work.

### E.0.7 MCAN

MCAN (Yu et al., 2019b) is a Transformer-based model that uses co-attention layers and a form of multi-hop reasoning to hone in on attended vision and language information. We use the model implementation by Yu et al. (2019a) to train the "small" model (6 layers).

### E.0.8 OSCAR+

OSCAR (Li et al., 2020) is a SOTA Transformer-based model that leverages pre-training on various V+L tasks and data sets. The subsequent release of new and elaborately trained visual features, known as VinVL (Zhang et al., 2021), further elevated its performance. We use this stronger version of OSCAR, called OSCAR+, in our evaluations. For training, we leverage the officially released pre-trained model and the VinVL features. Fine-tuning is done on GQA's balanced val set according to instructions accompanying the official release.

Note that we included results of UpDn (named "UpDn*", last line in Table 2) trained with these stronger VinVL features, in accordance with our recommendation in the Limitation section (§6) for new visual features.