# OpenReview forum: "Measuring Faithful and Plausible Visual Grounding in VQA"
_EMNLP/2023/Conference — EMNLP 2023 Findings_

### Official Review · Reviewer_bzSZ · 2023-08-04

**Soundness:** 2

**Excitement:**

2: Mediocre: This paper makes marginal contributions (vs non-contemporaneous work), so I would rather not see it in the conference.

**Paper Topic And Main Contributions:**

The paper introduces a new metric for evaluating the visual grounding abilities of VQA models, specifically focusing on the model's faithful and plausible reliance on relevant image regions. The paper provides experiments on multiple models with the proposed metric, which shows the effectiveness of the proposed metric.

**Reasons To Accept:**

- A metric to quantify a model’s faithful reliance on plausibly relevant image regions is proposed and can add value to the VQA research community.

**Reasons To Reject:**

- Lack of novelty. Several metrics for assessing visual grounding capabilities in VQA tasks already exist. The research questions explored in this paper may be limited in their degree of innovation compared to prior work. For example, Shrestha et al. (2020) also aimed to quantify VG capabilities based on relevant image regions.
- Lack of comparison with existing metrics, especially from the perspective of faithfulness and plausibility—the core characteristics of the proposed metric. While Table 1 provides some experimental results compared with existing metrics, a detailed comparison that focuses on these key characteristics would strengthen the paper's contribution and better highlight the advantages of the proposed approach.


**Reproducibility:**

4: Could mostly reproduce the results, but there may be some variation because of sample variance or minor variations in their interpretation of the protocol or method.

**Reviewer Confidence:**

3: Pretty sure, but there's a chance I missed something. Although I have a good feel for this area in general, I did not carefully check the paper's details, e.g., the math, experimental design, or novelty.

---

> ### Author Rebuttal · Authors · 2023-08-28
>
> We thank Reviewer bzSZ (hereafter “R3”) for their feedback.
>
> **Reject1:**
>
> R3 expresses concerns about the novelty of FPVG as a VG metric and points to Shrestha et al. (2020) as supporting reference.
>
> We strongly disagree with R3’s assessment in this point. For context: the metric used in Shrestha et al. (2020) relies on GradCAM measurements for calculating a ranking correlation between a model’s object rankings and the rankings based on object relevance annotations. GradCAM has been found to be among the least faithful measurement methods in the context of VQA’s UpDn model in Ying et al. (2022). FPVG on the other hand is shown in the paper to exhibit strong evidence of faithfulness (Sec 3.3). Moreover, the two methods are based on fundamentally different approaches to determine VG, with FPVG relying on feature manipulation and Shrestha et al. (2020) relying on gradient measurements. Essentially, the only thing the two methods have in common is their goal of measuring VG w.r.t. relevant image regions. Given these fundamental differences, it appears that R3’s expectation of novelty is that two methods mustn’t share the same goal* -- which is an extraordinarily restrictive opinion to maintain.
>
> *Indeed, R3 clearly states this opinion in their reasoning to reject the paper, namely “several metrics for assessing visual grounding capabilities in VQA tasks already exist”.
>
>
> **Reject2:**
>
> R3 states that the paper lacks detailed comparisons of faithfulness in existing metrics.
>
> R3 argues a moot point here and we’ll explain why. We’d like to clarify, that the paper’s goal is to introduce a new VG metric and show that it 1) measures VG in VQA in a “faithful” and “plausible” manner, and 2) is reliable and versatile to use in practice. These goals do not require the paper to provide a large-scale empirical study that ranks VG methods by faithfulness in a “detailed comparison”. Instead, the paper can (and does) accomplish the two listed goals as follows:
>
> 1. The paper addresses (1) by validating FPVG’s faithfulness property with three fundamentally different methods, two of which (LOO, Attention) have been found to provide the most faithful feature relevance determination in this particular experimental setting (see Ying et al. (2022)). The used methods each represent one of the three major categories of existing VG metrics, as mentioned in L313ff (incl. Shrestha et al. (2020)’s GradCAM based metric).
>
> 2. The paper addresses (2) by providing extensive evaluations across a large number of different VQA model architectures in both ID and OOD scenarios, hence demonstrating FPVG’s high utility in practical evaluations that other VG metrics might lack (see L75ff, L134ff, L148ff for problems when applying various other metrics). Experimental observations of FPVG consistently align with various theoretical assumptions made about VG in VQA context, such as VG’s role in OOD scenarios (Sec 4.4) and VG strength of certain model architectures like VLR (Table 2 and 3). The consistency of these results speak for FPVG’s reliability. The demonstrated applicability of FPVG to a wide variety of model architectures highlights its versatility.
>
> In summary, the paper provides adequate and sufficient support for its claims.

---

### Official Review · Reviewer_9Aq7 · 2023-08-06

**Typos Grammar Style And Presentation Improvements:** See some points in the weaknesses sec…
**Soundness:** 4

**Excitement:**

4: Strong: This paper deepens the understanding of some phenomenon or lowers the barriers to an existing research direction.

**Paper Topic And Main Contributions:**

Authors propose an evaluation paradigm of measuring faithful visual grounding in VQA. They propose that if an image is cropped to the relevant regions, the answer of a VQA system should remain unchanged compared to the full image. However, if the image is cropped to irrelevant regions, it should waver from the original answer. They make a metric based on this intuition and show that it is suggestive of OOD performance of VQA models.

**Questions For The Authors:**

- Does FPVG+ col in Table 3 mean the accuracy for the FVVG cases where the conditions (no change for relevant + change for irrelevant) are met?
- Why does Table 3 only have FPVG+ cases and not the - ones as well?
- See more questions in the weakness section.

**Reasons To Accept:**

This direction of measuring how much the object grounding is actually being used in the model predictions is important in evaluating models and making sure they don’t rely on biases.
The fact that a model answer should not change when cropped to relevant region of the image is a clean and simple way to make sure the model is consistent and grounded.
They show that FPVG and accuracy do not correlate with each other and hence, it provides additional information.
They show that the correct-to-incorrect ratio drops significantly for out-of-domain questions on the cases that their metric deems as bad cases vs good cases, showing that their metric is suggestive of OOD performance.

**Reasons To Reject:**

- I agreed with the authors on the their first premise - when only the relevant part of the image is cropped, the answer shouldn’t change.
However, I do not agree with the second part. When cropped to the irrelevant parts, I do not agree that the answer *has* to change. It may or may not change. This makes the metric overly harsh and might penalize a model unnecessarily. It might be nice to see if the second part of the metric is required for it to correlate well with OOD performance etc.

- The writing and flow could be improved. I am not clear why the FPVG has to be decomposed into 4 cases. Not sure what the differentiation of VQA answer correct cases and incorrect cases is giving us. It seems like all the analysis and results explained are based on the + and -. It is unclear in Table 2 what trend to look for in the 8th and 10th columns.

- The result presentation can be simplified. It is hard to know which number to compare with which. It’s unclear what Table 3 is actually showing, Is it showing the FPVG+ metric number or the accuracy on the cases where the FPVG conditions are met? Which numbers are we actually comparing here?  Is it that the OOD for FPVG column correlates with the OOD accuracy? Explicitly pointing these out in the result section would help.

The results look promising. If the authors can 1) justify that having the requirement of answer changing when irrelevant is required or at least not hurtful (even if overly harsh) to ranking better models higher, and 2) improve and simplify the presentation of results (make key take away points as headings and explicitly point to numbers using row/col numbers if required in the Tables we should be looking to support the claim), I will be willing to increase my rating.

**[EDIT] I have upgraded my rating due to the inclusion of the experiments on how the metric correlates with OOD performance if the requirement for the answer to change for irrelevant input is removed. This is interesting, and I hope to see some discussion and intuition behind why this is the case.**

**Reproducibility:**

4: Could mostly reproduce the results, but there may be some variation because of sample variance or minor variations in their interpretation of the protocol or method.

**Reviewer Confidence:**

4: Quite sure. I tried to check the important points carefully. It's unlikely, though conceivable, that I missed something that should affect my ratings.

---

> ### Author Rebuttal · Authors · 2023-08-28
>
> We thank Reviewer 9Aq7 (hereafter “R2”) for their feedback. We particularly value R2’s clearly stated requirements for a rating improvement and hope we satisfied these requirements below.
>
> **Reject1:**
> One motivation for considering an answer change when testing with irrelevant parts is given in L271ff, namely that it uncovers cases where the model is simply indifferent to visual input entirely. This indifference to visual input is a major (language) bias problem in VQA and hence it’s important to have a mechanism that can identify these cases.
>
> Nevertheless, for R2’s consideration, we modified FPVG to only consider answer changes when testing with relevant objects as per R2’s request. Average metric results for the ID/OOD tests are as follows:
>
> | Model |  ID | OOD |
> | --------| -------| ------|
> |UpDn | 0.77 | 0.77 |
> |HINT| 0.75 | 0.75|
> |VisFIS| 0.82 | 0.80|
> |MAC| 0.73 | 0.73|
> |MMN| 0.62 | 0.58|
> |VLR| 0.79 | 0.79|
>
> Results of R2’s proposed metric appear to be less reasonable than FPVG. E.g. VLR, which achieves by far the best FPVG and OOD performance is now ranked behind VisFIS and close to UpDn. MMN, which had the best OOD performance among classification-based models (and was ranked third in FPVG) is now ranked last by a large margin. Based on the architectural properties of these models, such rankings would be very surprising.
>
> We also investigated the c2i ratios for R2’s metric in the OOD scenario (not posted here, but can be shared if required). Here, we observe opposite trends to the ones shown in the paper. In particular, these new results suggest that well-grounded questions (as per R2’s proposed metric) are much more prone to producing wrong answers in OOD vs ID than badly-grounded questions. This doesn’t align with any reasonable expectation for a model’s OOD behavior and we think it again points to problems with R2’s proposed metric.
>
> These additional results on top of the motivation / explanation in the paper (L271ff) emphasize the necessity of including tests with irrelevant objects in FPVG. We hope this satisfies R2's concerns.
>
>
> **Reject2:**
> - FPVG’s 4 cases: The subdivision of FPVG by answer correctness facilitates in-depth analysis. E.g., they provide the basis for the analysis in Sec. 4.4 (c2i ratios). We find these additional categories helpful and informative in our research (also beyond this paper) and hence found it important to explicitly point them out.
>
> - Table 2 trends in 8th/10th column: Since there is no intuitively sensible ranking for the mentioned categories under the FPVG motivation, we made a conscious decision not to assign any ranking directive here. That does not however mean that these two categories are irrelevant, as they can be informative for different types of analyses (as shown in Sec. 4.4). We'll explicitly point this out in the final revision.
>
>
> **Reject3:**
> Table 3 shows both Accuracy and FPVG+, both over the same set of questions. The two FPVG+ columns in Table 3 list actual FPVG+ values and not accuracy. Looking at Table 3, we find evidence that FPVG+ does not linearly predict OOD accuracy. We discuss this and identify reasons for it in Sec 4.4, L558ff. We'd like to point out that some evidence was presented in Ying et al. (2022) to suggest that VG, or RRR metrics in general, might not be an ideal predictor for OOD accuracy. Still, our paper shows: while using FPVG+ as a straight-up predictor for accuracy might be problematic, FPVG-based analysis can be used to explain a model’s obstacles in OOD settings, as well as the role VG plays in VQA models for achieving OOD performances on par with ID settings.
>
>
>
> **Reject4 (without bullet point):**
> We’ll look into adding additional highlighting for the take-away points, as requested by R2. Other points mentioned in this paragraph were addressed above.
>
> **Question1:**
> See Reject3 above. We'll add clarification to Table 3's caption in the revision.
>
> **Question2:**
> FPVG- is not listed because it can be calculated based on the listed FPVG+ values (see Eq.6: FPVG- = 1 - FPVG+).

---

### Official Review · Reviewer_fV5W · 2023-08-07

**Typos Grammar Style And Presentation Improvements:** Figure 2 was difficult to follow. I w…
**Soundness:** 4

**Excitement:**

4: Strong: This paper deepens the understanding of some phenomenon or lowers the barriers to an existing research direction.

**Paper Topic And Main Contributions:**

The authors propose a new Visual Grounding (VG) metric for VQA (Visual Question Answering) that captures (i) if a model identifies question relevant objects in the scene and (ii) actually relies on the information contained in the relevant objects when producing the answer. The authors name their metric Faithful and Plausible Visual Grounding (FPVG) and evaluate it on several VQA architectures. They also provide the code for the calculations.

**Reasons To Accept:**

This paper is well written. The problem and solution is well articulated and backed up by solid results. The metric FPVG is simple and intuitive. The research community can widely adopt this metric to measure VG in VQA tasks.

**Reasons To Reject:**

None

**Reproducibility:**

4: Could mostly reproduce the results, but there may be some variation because of sample variance or minor variations in their interpretation of the protocol or method.

**Reviewer Confidence:**

3: Pretty sure, but there's a chance I missed something. Although I have a good feel for this area in general, I did not carefully check the paper's details, e.g., the math, experimental design, or novelty.

---

> ### Author Rebuttal · Authors · 2023-08-28
>
> We thank Reviewer fV5W (hereafter “R1”) for their feedback.
>
> R1 recommends a more detailed description of Figure 2. We’ll gladly add this to the final version of the paper.

---

### Meta-Review · Area_Chair_Jc4E · 2023-09-19

**Recommendation:** 4

**Metareview:**

This paper proposes a new Visual Grounding (VG) metric for VQA aiming to balance both faithfulness and plausibility, i.e. (i) if a model identifies question relevant objects in the scene and (ii) actually relies on the information contained in the relevant objects when producing the answer.

The majority of reviewers felt that the paper is well written with conclusions backed up by experiments. The new metrics give new insights that are not captured by existing ones, and can inspire the design and evaluation of new methods.

There were some concerns over writing and structuring of the paper, as well as concerns regarding some design decisions in the metric by reviewer 9Aq7, which the authors adequately addressed during discussion. Reviewer bzSZ raised several issues as well, in particular that the metric overlaps slightly with other investigations regarding metrics for grounding, which deserves additional comparisons. I believe the authors gave satisfactory responses regarding the differences in their method, but I would still strongly suggest adding a discussion wrt these other metrics. How can researchers decide which of these metrics to use or optimize their models for? Can we study the similarities and differences between these metrics systematically?

---

### Decision · Program_Chairs · 2023-10-07

**Decision:**

Accept-Findings

**Comment:**

This paper proposes a new Visual Grounding (VG) metric for VQA aiming to balance both faithfulness and plausibility, i.e. (i) if a model identifies question relevant objects in the scene and (ii) actually relies on the information contained in the relevant objects when producing the answer.

The majority of reviewers felt that the paper is well written with conclusions backed up by experiments. The new metrics give new insights that are not captured by existing ones, and can inspire the design and evaluation of new methods.

There were some concerns over writing and structuring of the paper, as well as concerns regarding some design decisions in the metric by reviewer 9Aq7, which the authors adequately addressed during discussion. Reviewer bzSZ raised several issues as well, in particular that the metric overlaps slightly with other investigations regarding metrics for grounding, which deserves additional comparisons. I believe the authors gave satisfactory responses regarding the differences in their method, but I would still strongly suggest adding a discussion wrt these other metrics. How can researchers decide which of these metrics to use or optimize their models for? Can we study the similarities and differences between these metrics systematically?